# Graph Meta Learning via Local Subgraphs

**Kexin Huang**
Harvard University
kexinhuang@hsph.harvard.edu

**Marinka Zitnik**
Harvard University
marinka@hms.harvard.edu

## Abstract

Prevailing methods for graphs require abundant label and edge information for learning. When data for a new task are scarce, meta learning can learn from prior experiences and form much-needed inductive biases for fast adaption to new tasks. Here, we introduce G-META, a novel meta-learning algorithm for graphs. G-META uses local subgraphs to transfer subgraph-specific information and learn transferable knowledge faster via meta gradients. G-META learns how to quickly adapt to a new task using only a handful of nodes or edges in the new task and does so by learning from data points in other graphs or related, albeit disjoint, label sets. G-META is theoretically justified as we show that the evidence for a prediction can be found in the local subgraph surrounding the target node or edge. Experiments on seven datasets and nine baseline methods show that G-META outperforms existing methods by up to 16.3%. Unlike previous methods, G-META successfully learns in challenging, few-shot learning settings that require generalization to completely new graphs and never-before-seen labels. Finally, G-META scales to large graphs, which we demonstrate on a new Tree-of-Life dataset comprising 1,840 graphs, a two-orders of magnitude increase in the number of graphs used in prior work.

## 1  Introduction

Graph Neural Networks (GNNs) have achieved remarkable results in domains such as recommender systems [56], molecular biology [65, 19], and knowledge graphs [49, 18]. Performance is typically evaluated after extensive training on datasets where majority of labels are available [52, 54]. In contrast, many problems require rapid learning from only a few labeled nodes or edges in the graph. Such flexible adaptation, known as meta learning, has been extensively studied for images and language, *e.g.*, [39, 51, 27]. However, meta learning on graphs has received considerably less research attention and has remained a problem beyond the reach of prevailing GNN models.

Meta learning on graphs generally refers to a scenario in which a model learns at two levels. In the first level, rapid learning occurs *within* a task. For example, when a GNN learns to classify nodes in a particular graph accurately. In the second level, this learning is guided by knowledge accumulated gradually *across* tasks to capture how the task structure changes across target domains [35, 4, 34]. A powerful GNN trained to meta-learn can quickly learn never-before-seen labels and relations using only a handful of labeled data points. As an example, a key problem in biology is to translate insights from non-human organisms (such as yeast, zebrafish, and mouse) to humans [66]. How to train a GNN to effectively meta-learn on a large number of incomplete and scarcely labeled protein-protein interaction (PPI) graphs from various organisms, transfer the accrued knowledge to humans, and use it to predict the roles of protein nodes in the human PPI graph? While this is a hard task, it is only an instance of a particular graph meta-learning problem (Figure 1C). In this problem, the GNN needs to learn on a large number of graphs, each scarcely labeled with a unique label set. Then, it needs to quickly adapt to a never-before-seen graph (*e.g.*, a PPI graph from a new organism) and never-before-seen labels (*e.g.*, newly discovered roles of proteins). Current methods are specialized techniques specifically designed for a particular problem and a particular task [63, 3, 24, 7, 53].

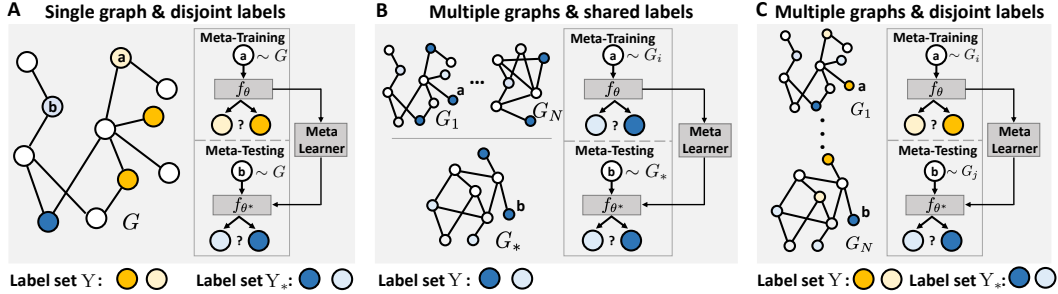

Figure 1: Graph meta-learning problems. **A.** Meta-learner classifies unseen label set by observing other label sets in the same graph. **B.** Meta-learner learns unseen graph by learning from other graphs with the same label set. **C.** Meta-learner classifies unseen label set by learning from other label sets across multiple graphs. Unlike existing methods, G-META solves all three problems and also works for link prediction (see Section 6.1).

While these methods provide a promising approach to meta learning in GNNs, their specific strategy does not scale well nor extend to other problems (Figure 1).

**Present work.** We introduce G-META,[1] an approach for meta learning on graphs (Figure 1). The core principle of G-META is to represent every node with a local subgraph and use subgraphs to train GNNs to meta-learn. Our theoretical analysis (Section 4) suggests that the evidence for a prediction can be found in the subgraph surrounding the target node or edge when using GNNs. In contrast to G-META, earlier techniques are trained to meta-learn on entire graphs. As we show theoretically and empirically, such methods are unlikely to succeed in few-shot learning settings when the labels are scarce and scattered around multiple graphs. Furthermore, previous methods capture the overall graph structure but at the loss of finer local structures. Besides, G-META's construction of local subgraphs gives local structural representations that enable direct structural similarity comparison using GNNs based on its connection to Weisfeiler-Lehman test [54, 60]. Further, structural similarity enables G-META to form the much-needed inductive bias via a metric-learning algorithm [37]. Moreover, local subgraphs also allow for effective feature propagation and label smoothing within a GNN.

(1) G-META is a general approach for a variety of meta learning problems on graph-structured data. While previous methods [63, 3] apply only to one graph meta-learning problem (Figure 1), G-META works for all of them (Appendix B). (2) G-META yields accurate predictors. We demonstrate G-META's performance on seven datasets and compare it against nine baselines. G-META considerably outperforms baselines by up to 16.3%. (3) G-META is scalable. By operating on subgraphs, G-META needs to examine only small graph neighborhoods. We show how G-META scales to large graphs by applying it to our new Tree-of-Life dataset comprising 1,840 graphs, a two-orders of magnitude increase in the number of graphs used in prior work.

## 2 Related Work

(1) Few-shot meta learning. Few-shot meta learning learns from prior experiences and transfers them to a new task using only a few labeled examples [46, 41]. Meta learning methods generally fall into three categories, model-based [14, 50, 34], metric-based [37, 38, 47], and optimization-based [32, 29, 12] techniques. (2) Meta learning for graphs. Recent studies incorporate graph-structured data into meta learning. Liu *et al.* [24] construct image category graphs to improve few-shot learning. Zhou *et al.* [62] use graph meta learning for fast network alignment. MetaR [7] and GMatching [53] use metric methods to generalize over new relations from a handful of associative relations in a knowledge graph. While these methods learn from a single graph, G-META can handle settings with many graphs and disjoint label sets. Further, Chauhan *et al.* [5] use a super-class prototypical network for graph classification. In contrast, G-META focuses on node classification and link prediction. In settings with single graphs and disjoint labels (Figure 1), Meta-GNN [63] uses gradient-based meta-learning for node classification. On a related note, GFL [55] focuses on settings with multiple graphs and shared labels across the graphs. Similarly, Meta-Graph [3] uses graph signature functions for few-shot link prediction across multiple graphs. In contrast, our G-META can be used for many more problems (Figure 1). Further, Meta-GNN operates a GNN on

an entire graph, whereas G-META extracts relevant local subgraphs first and then trains a GNN on each subgraph individually. In Meta-GNN, a task is defined as a batch of node embeddings, whereas in G-META, a task is given by a batch of subgraphs. This difference allows for rapid adaptation to new tasks, improves scalability, and applies broadly to meta learning problems. In summary, G-META works for all problems in Figure 1 and applies to node classification and link prediction. (3) Subgraphs and GNNs. The ability to model subgraph structure is vital for numerous graph tasks, *e.g.*, [45, 9, 58, 40]. For example, Patchy-San [30] uses local receptive fields to extract useful features from graphs. Ego-CNN uses ego graphs to find critical graph structures [43]. SEAL [61] develops theory showing that enclosing subgraphs capture graph heuristics, which we extend to GNNs here. Cluster-GCN [8] and GraphSAINT [59] use subgraphs to improve GNN scalability. G-META is the first approach to use subgraphs for meta learning.

# 3 Background and Problem Formulation

Let $\mathcal{G} = \{G_1, \ldots, G_N\}$ denote $N$ graphs. For each graph $G = (\mathcal{V}, \mathcal{E}, \mathbf{X})$, $\mathcal{V}$ is a set of nodes, $\mathcal{E}$ is a set of edges, and $\mathbf{X} = \{\mathbf{x}_1, \ldots, \mathbf{x}_n\}$ is a set of attribute vectors, where $\mathbf{x}_u \in \mathbb{R}^d$ is a $d$-dimensional feature vector for node $u \in \mathcal{V}$. We denote $\mathcal{Y} = \{Y_1, \ldots, Y_M\}$ as a set of $M$ distinct labels. We use Y to denote a set of labels selected from $\mathcal{Y}$. G-META's core principle is to represent nodes with local subgraphs and then use subgraphs to transfer knowledge across tasks, graphs, and sets of labels. We use $\mathcal{S}$ to denote local subgraphs for nodes, $\mathcal{S} = \{S_1, \ldots, S_n\}$. Node classification aims to specify a GNN $f_\theta : \mathcal{S} \mapsto \{1, \ldots, |\mathrm{Y}|\}$ that can accurately map node $u$'s local subgraph $S_u$ to labels in Y given only a handful of labeled nodes.

**Background on graph neural networks.** GNNs learn compact representations (embeddings) that capture network structure and node features. A GNN generates outputs through a series of propagation layers [13], where propagation at layer $l$ consists of the following three steps: (1) Neural message passing. GNN computes a message $\mathbf{m}_{uv}^{(l)} = \mathrm{MSG}(\mathbf{h}_u^{(l-1)}, \mathbf{h}_v^{(l-1)})$ for every linked nodes $u, v$ based on their embeddings from the previous layer $\mathbf{h}_u^{(l-1)}$ and $\mathbf{h}_v^{(l-1)}$. (2) Neighborhood aggregation. The messages between node $u$ and its neighbors $\mathcal{N}_u$ are aggregated as $\hat{\mathbf{m}}_u^{(l)} = \mathrm{AGG}(\mathbf{m}_{uv}^{(l)} | v \in \mathcal{N}_u)$. (3) Update. Finally, GNN uses a non-linear function to update node embeddings as $\mathbf{h}_u^{(l)} = \mathrm{UPD}(\hat{\mathbf{m}}_u^{(l)}, \mathbf{h}_u^{(l-1)})$ using the aggregated message and the embedding from the previous layer.

**Background on meta learning.** In meta learning, we have a meta-set $\mathscr{D}$ that consists of $\mathscr{D}_{\mathrm{train}}, \mathscr{D}_{\mathrm{val}}, \mathscr{D}_{\mathrm{test}}$. This meta-set consists of many tasks. Each task $\mathcal{T}_i \in \mathscr{D}$ can be divided into $\mathcal{T}_i^{\mathrm{support}}$ and $\mathcal{T}_i^{\mathrm{query}}$. $\mathcal{T}_i^{\mathrm{support}}$ has $\mathrm{K}_{\mathrm{support}}$ labeled data points in each label for learning and $\mathcal{T}_i^{\mathrm{query}}$ has $\mathrm{K}_{\mathrm{query}}$ data points in each label for evaluation. The size of the label set for the meta-set $|\mathrm{Y}|$ is N. It is also called N-ways $\mathrm{K}_{\mathrm{support}}$-shots learning problem. During meta-training, for $\mathcal{T}_i$ from $\mathscr{D}_{\mathrm{train}}$, the model first learns from $\mathcal{T}_i^{\mathrm{support}}$ and then evaluates on $\mathcal{T}_i^{\mathrm{query}}$ to see how well the model performs on that task. The goal of Model-Agnostic Meta-Learning (MAML) [12] is to obtain a parameter initialization $\theta_*$ that can adapt to unseen tasks quickly, such as $\mathscr{D}_{\mathrm{test}}$, using gradients information learnt during meta-training. Hyperparameters are tuned via $\mathscr{D}_{\mathrm{val}}$.

## 3.1 G-META: Problem Formulation

G-META is designed for three fundamentally different meta-learning problems (Figure 1). "Shared labels" refers to the situation where every task shares the same label set Y. "Disjoint labels" refers to the situation where label sets for tasks $i$ and $j$ are disjoint, *i.e.*, $\mathrm{Y}_i \cap \mathrm{Y}_j = \emptyset$. Each data point in a task $\mathcal{T}_i$ is a local subgraph $S_u$, along with its associated label $Y_u$. G-META aims to adapt to a new task $\mathcal{T}_* \sim p(\mathcal{T})$ for which only a handful examples are available after observing related tasks $\mathcal{T}_i \sim p(\mathcal{T})$.

**Graph meta-learning problem 1: Single Graph and Disjoint Labels.** We have graph $G$ and a distribution of label sets $p(\mathrm{Y}|G)$. The goal is to adapt to an unseen label set $\mathrm{Y}_* \sim p(\mathrm{Y}|G)$ by learning from tasks with other label sets $\mathrm{Y}_i \sim p(\mathrm{Y}|G)$, where $\mathrm{Y}_i \cap \mathrm{Y}_* = \emptyset$ for every label set $\mathrm{Y}_i$.
**Graph meta-learning problem 2: Multiple Graphs and Shared Labels.** We have a distribution of graphs $p(G)$ and one label set Y. The goal is to learn from graph $G_j \sim p(G)$ and quickly adapt to an unseen graph $G_* \sim p(G)$, where $G_j$ and $G_*$ are disjoint. All tasks share the same labels.

**Graph meta-learning problem 3: Multiple Graphs and Disjoint Labels.** We have a distribution of label sets $p(Y|\mathcal{G})$ conditioned on multiple graphs $\mathcal{G}$. Each task has its own label set $Y_i$ but the same label set can appear in multiple graphs. The goal is to adapt to an unseen label set $Y_* \sim p(Y|\mathcal{G})$ by learning from a disjoint label set $Y_i \sim p(Y|\mathcal{G})$, where $Y_i \cap Y_* = \emptyset$.

# 4  Local Subgraphs and Theoretical Motivation for G-META

We start by describing how to construct local subgraphs in G-META. We then provide a theoretical justification showing that local subgraphs preserve useful information from the entire graph. We then argue how subgraphs enable G-META to capture sufficient information on the graph structure, node features, and labels, and use that information for graph meta-learning.

For node $u$, a local subgraph is defined as a subgraph $S_u = (\mathcal{V}^u, \mathcal{E}^u, \mathbf{X}^u)$ induced from a set of nodes $\{v|d(u,v) \leq h\}$, where $d(u,v)$ is the shortest path distance between node $u$ and $v$, and $h$ defines the neighborhood size. In a meta-task, subgraphs are sampled from graphs or label sets, depending on the graph meta-learning problems defined in Section 3. We then use GNNs to encode the local subgraphs. However, one straightforward question raised is if this subgraph loses information by excluding nodes outside of it. Here, we show in theory that applying GNN on the local subgraph preserve useful information compared to using GNN on the entire graph.

**Preliminaries and definitions.** Next, we use GCN [22] as an exemplar GNN to understand how nodes influence each other during neural message passing. The assumptions are based on [48] and are detailed in Appendix C. We need the following definitions. (1) Node influence $I_{u,v}$ of $v$ on $u$ in the final GNN output is: $I_{u,v} = \|\partial \mathbf{x}_u^{(\infty)} / \partial \mathbf{x}_v^{(\infty)}\|$, where the norm is any subordinate norm and the Jacobian measures how a change in $v$ translates to a change in $u$ [48]. (2) Graph influence $I_G$ on $u$ is: $I_G(u) = \|[I_{u,v_1}, \ldots, I_{u,v_n}]\|_1$, where $[I_{u,v_1}, \ldots, I_{u,v_n}]$ is a vector representing the influence of other nodes on $u$. (3) Graph influence loss $R_h$ is defined as: $R_h(u) = I_G(u) - I_{S_u}(u)$, where $I_G(u)$ is the influence of entire graph $G$, and $I_{S_u}(u)$ is the influence of local subgraph $S_u$. Next, we show how influence spreads between nodes depending on how far the nodes are from each other in a graph.

**Theorem 1 (Decaying Property of Node Influence).** *Let $t$ be a path between node $u$ and node $v$ and let $D_{\mathrm{GM}}^t$ be a geometric mean of node degrees occurring on path $t$. Let $D_{\mathrm{GM}}^{t_*} = \min_t \{D_{\mathrm{GM}}^t\}$ and $h_* = d(u,v)$. Consider the node influence $I_{u,v}$ from $v$ to $u$. Then, $I_{u,v} \leq C/(D_{\mathrm{GM}}^{t_*})^{h_*}$.*

The proof is deferred to Appendix C. Theorem 1 states that the influence of a node $v$ on node $u$ decays exponentially as their distance $h_*$ increases. A side theoretical finding is that the influence is significantly decided by the accumulated node degrees of the paths between two nodes. In other words, if the paths are straight lines of nodes (low accumulated node degrees), then the node influence is high. Otherwise, if the paths consist of numerous connections to other nodes (high accumulated node degrees), the node influence is minimum. Intuitively, high accumulated node degrees can bring complicated messages along the paths, which dampen each individual node influence whereas low degrees paths can pass the message directly to the target node. Since real-world graphs are usually complicated graphs with relatively high node degrees, the node influence will be considerably low. We then show that for a node $u$, operating GNN on a local subgraph does not lose useful information than operating GNN on the entire graph.

**Theorem 2 (Local Subgraph Preservation Property).** *Let $S_u$ be a local subgraph for node $u$ with neighborhood size $h$. Let node $v$ be defined as: $v = \mathrm{argmax}_w(\{I_{u,w}|w \in \mathcal{V} \setminus \mathcal{V}^u\})$. Let $\bar{t}$ be a path between $u$ and $v$ and let $D_{\mathrm{GM}}^{\bar{t}}$ be a geometric mean of node degrees occurring on path $\bar{t}$. Let $D_{\mathrm{GM}}^{\bar{t}_*} = \min_{\bar{t}}\{D_{\mathrm{GM}}^{\bar{t}}\}$. The following holds: $R_h(u) \leq C/(D_{\mathrm{GM}}^{\bar{t}_*})^{h+1}$.*

The proof is deferred to Appendix D. Theorem 2 states that the graph influence loss is bounded by an exponentially decaying term as $h$ increases. In other words, the local subgraph formulation is an $h$-th order approximation of applying GNN for the entire graph.

**Local subgraphs enable few-shot meta-learning.** Equipped with the theoretical justification, we now describe how the local subgraph allows G-META to do few-shot meta-learning. The construction of local subgraphs captures the following. (1) Structures. Graph structures present an alternative source of strong signal for prediction [9, 45], especially when node labels are scarce. The GNN representations cannot fully capture large graphs structure because they are too complicated [3, 54]. However, they can learn representations that capture the structure of small graphs, such as our

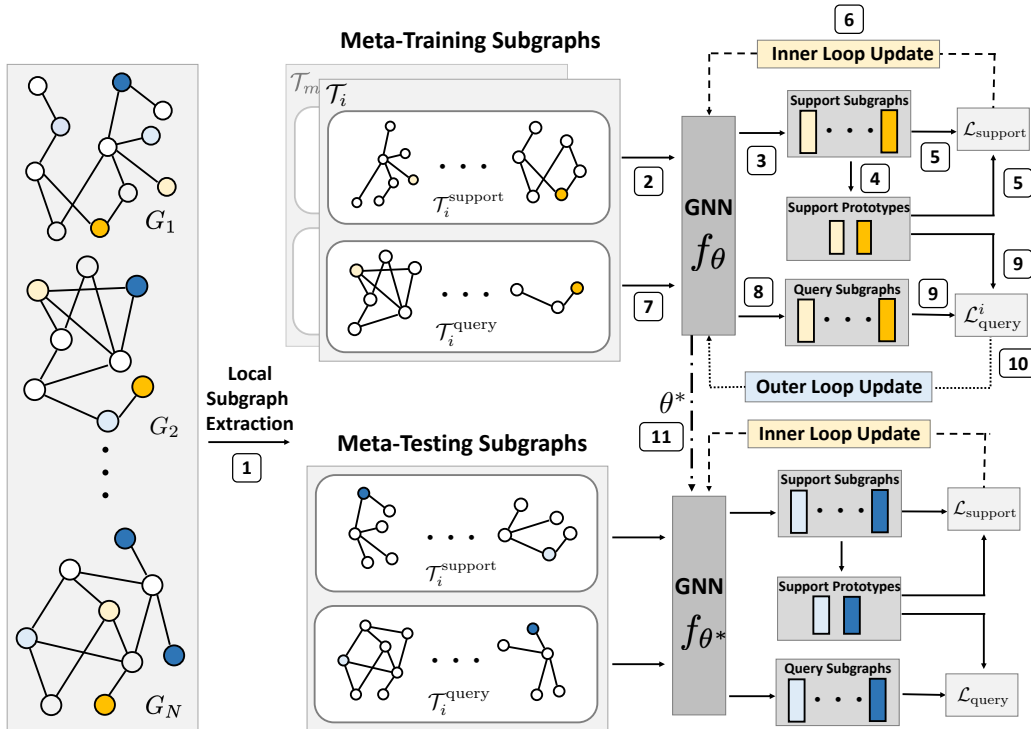

**Figure 2:** (1) We first construct a batch of $m$ meta-training tasks and extract local subgraphs on the fly for nodes in the meta-tasks. For each task $\mathcal{T}_i$, (2) subgraphs from the support set are mini-batched and are fed into a GNN parameterized by $\theta$. (3) The support set embeddings using the centroid nodes are generated, and (4) the prototypes are computed from the support centroid embeddings. Then, (5) the support set loss $\mathcal{L}_{\text{support}}$ is computed, and (6) back-propagates to update the GNN parameter. (7) $\mathcal{T}_i^{\text{query}}$ subgraphs then feed into the updated GNN to (8) generate query centroid embeddings. (9) Using the support prototypes and the query embeddings, the query loss $\mathcal{L}_{\text{query}}^i$ for task $\mathcal{T}_i$ is computed. Steps (2-9) are repeated for $\eta$ update steps. The same process repeats for the other $m$ sampled tasks, starting from the same GNN $f_\theta$. (10) The last update step's query loss from all the tasks are summed up and used to update $\theta$. Then, another batch of tasks are sampled, and step (1-10) are repeated. Then, for meta-testing tasks, steps (1-9) are repeated with the GNN using the meta-learned parameter $\theta_*$, which enables generalization over unseen tasks. See Algorithm 1 (Appendix E).

local subgraphs, as is evidenced by the connection to the Weisfeiler-Lehman test [54, 60]. Hence, subgraphs enable G-META to capture structrual node information. (2) Features. Local subgraphs preserve useful information, as indicated by the theorems above. (3) Labels. When only a handful of nodes are labeled, it is challenging to efficiently propagate the labels through the entire graph [64, 23]. Metric-learning methods [37] learn a task-specific metric to classify query set data using the closest point from the support set. It has been proved as an effective inductive bias [37, 42]. Equipped with subgraph representations that capture both structure and feature information, G-META uses metric-learning by comparing the query subgraph embedding to the support subgraph embedding. As such, it circumvents the problem of having too little label information for effective propagation.

## 5 G-META: Meta Learning via Local Subgraphs

G-META (Figure 2) is an approach for meta-learning on graphs. Building on theoretical motivation from Section 4, G-META first constructs local subgraphs. It then uses a GNN encoder to generate embeddings for subgraphs. Finally, it uses prototypical loss for inductive bias and MAML for knowledge transfer across graphs and labels. The overview is in Algorithm 1 (Appendix E).

**Neural encoding of subgraphs.** In each meta-task, we first construct a subgraph $S_u$ for each node $u$. While we use $h$-hops neighbors to construct subgraphs, other subgraph extraction algorithms, e.g., [6, 11] can be considered. We then feed each subgraph $S_u$ into a $h$-layer GNN to obtain an embedding for every node in the subgraph. Here, $h$ is set to the size of subgraph neighborhood. The

centroid node $u$'s embedding is used to represent the subgraph $\mathbf{h}_u = \text{Centroid}(\text{GNN}(S_u))$. Note that a centroid node embedding is a particular instantiation of our framework. One can consider alternative subgraph representations, such as subgraph neural networks [1, 57] or readout functions specified over nodes in a subgraph [54, 5]. Notably, local subgraphs in our study are different from computation graphs [58]. Local subgraphs are not used for neural message passing; instead we use them for meta learning. Further, our framework does not constrain subgraphs to $h$-hop neighborhoods or subgraph neural encodings to centroid embeddings.

**Prototypical loss.** After we obtain subgraph representations, we leverage inductive bias between representations and labels to circumvent the issue of limited label information in few-shot settings. For each label $k$, we take the mean over support set subgraph embeddings to obtain a prototype $\mathbf{c}_k$ as: $\mathbf{c}_k = 1/N_k \sum_{y_j=k} \mathbf{h}_j$. The prototype $\mathbf{c}_k$ serves as a landmark for label $k$. Then, for each local subgraph $S_u$ that exists in either support or query set, a class distribution vector $\mathbf{p}$ is calculated via the Euclidean distance between support prototypes for each class and centroid embeddings as: $\mathbf{p}_k = (\exp(-\|\mathbf{h}_u - \mathbf{c}_k\|))/(\sum_{\hat{k}} \exp(-\|\mathbf{h}_u - \mathbf{c}_{\hat{k}}\|))$. Finally, we use class distribution vectors from local subgraphs to optimize a cross-entropy loss as follows: $\text{L}(\mathbf{p}, \mathbf{y}) = \sum_j \mathbf{y}_j \log \mathbf{p}_j$, where $\mathbf{y}$ indicates a true label's one-hot encoding.

**Optimization-based meta-learning.** To transfer the structural knowledge across graphs and labels, we use MAML, an optimization-based meta-learning approach. We break the node's dependency on a graph by framing nodes into independent local subgraph classification tasks. It allows direct adaptation to MAML since individual subgraphs can be considered as an individual image in the classic few-shot meta-learning setups. More specifically, we first sample a batch of tasks, where each task consists of a set of subgraphs. During meta-training inner loop, we perform the regular stochastic gradient descent on the support loss for each task $\mathcal{T}_i$: $\theta_j = \theta_{j-1} - \alpha \nabla \mathcal{L}_{\text{support}}$. The updated parameter is then evaluated using the query set, and the query loss for task $i$ is recorded as $\mathcal{L}_{\text{query}}^i$. The above steps are repeated for $\eta$ update steps. Then, the $\mathcal{L}_{\text{query}}^i$ from the last update step is summed up across the batch of tasks, and then we perform a meta-update step: $\theta = \theta - \beta \nabla \sum_i \mathcal{L}_{\text{query}}^i$. Next, new tasks batch are sampled, and the same iteration is applied on the meta-updated $\theta$. During meta-testing, the same procedure above is applied using the final meta-updated parameter $\theta_*$. $\theta_*$ is learned from knowledge across meta-training tasks and is the optimal parameter to adapt to unseen tasks quickly.

**Attractive properties of G-META.** (1) Scalability: G-META operates on a mini-batch of local subgraphs where the subgraph size and the batch size (few-shots) are both small. This allows fast computation and low memory requirement because G-META's aggregation field is smaller than that of previous methods that operate on entire graphs [3, 63]. We further increase scalability by sub-sampling subgraphs with more than 1,000 nodes. Also note that graph few-shot learning does not evaluate all the samples in the graph since few-shot learning means majority of labels do not exist. Thus, each mini-batch consists only of a few samples with labels (e.g., 9 in 3-way 3-shot learning), which are quick operations. (2) Inductive learning: Since G-META operates on different subgraphs in each GNN encoding, it forces inductiveness over unseen subgraphs. This inductiveness is crucial for few-shot learning, where the trained model needs to adapt to unseen nodes. Inductive learning also allows for knowledge transfer from meta-training subgraphs to meta-testing subgraphs. (3) Over-smoothing regularization: One limitation of GNN is that connected nodes become increasingly similar after multiple iterations of propagation on the same graph. In contrast, each iteration for G-META consists of a batch of different subgraphs with various structures, sizes and nodes, where each subgraph is fed into the GNN individually. This prevents GNN from over-smoothing on the structure of a single graph. (4) Few-shot learning: G-META needs only a tiny number of labeled nodes for successful learning, as demonstrated in experiments. This property is in contrast with prevailing GNNs, which require a large fraction of labeled nodes to propagate neural messages in the graph successfully. (5) Broad applicability: G-META applies to many graph meta-learning problems (Figure 1) whereas previous methods apply to at most one [3, 63]. Unlike earlier methods, G-META works for node classification and few-shot link prediction (*i.e.*, via local subgraphs for a pair of nodes).

## 6   Experiments

**Synthetic datasets.** We have two synthetic datasets whose labels depend on nodes' structural roles [16], which we use to confirm that G-META captures local graph structure (Table 1). (1) Cycle:

**Table 1:** Dataset statistics. Fold-PPI and Tree-of-Life are new datasets introduced in this study.

| Dataset | Task | # Graphs | # Nodes | # Edges | # Features | # Labels |
|---|---|---|---|---|---|---|
| Synthetic Cycle | Node | 10 | 11,476 | 19,687 | N/A | 17 |
| Synthetic BA | Node | 10 | 2,000 | 7,647 | N/A | 10 |
| ogbn-arxiv | Node | 1 | 169,343 | 1,166,243 | 128 | 40 |
| Tissue-PPI | Node | 24 | 51,194 | 1,350,412 | 50 | 10 |
| FirstMM-DB | Link | 41 | 56,468 | 126,024 | 5 | 2 |
| Fold-PPI | Node | 144 | 274,606 | 3,666,563 | 512 | 29 |
| Tree-of-Life | Link | 1,840 | 1,450,633 | 8,762,166 | N/A | 2 |

we use a cycle basis graph and attach a distribution of shapes: House, Star, Diamond, Fan [9]. The label of each node is the structural role define by the shape. We also add random edges as noise. In the multiple graphs problem, each graph is with varying distribution of number of shapes. (2) BA: to model local structural information under a more realistic homophily graph, we construct a Barabási-Albert (BA) graph and then plant different shapes to the graph. Then, we compute the Graphlet Distribution Vector [31] for each node, which characterizes the local graph structures and then we apply spectral clustering on this vector to generate the labels. For multiple graphs problem, a varying distribution of numbers of shapes are used to plant each BA graph. Details are in Appendix F.

**Real-world datasets and new meta-learning datasets.** We use three real world datasets for node classification and two for link prediction to evaluate G-META (Table 1). (1) ogbn-arxiv is a CS citation network, where features are titles, and labels are the subject areas [17]. (2) Tissue-PPI consists of 24 protein-protein interaction networks from different tissues, where features are gene signatures and labels are gene ontology functions [67, 15]. (3) Fold-PPI is a novel dataset, which we constructed for the multiple graph and disjoint label problem. It has 144 tissue PPI networks [67], and the labels are protein structures defined in SCOP database [2]. The features are conjoint triad protein descriptor [36]. We screen fold groups that have more than 9 unique proteins across the networks. It results in 29 unique labels. Like many real-world graphs, in Fold-PPI, the majority of the nodes do not have associated labels. This also shows the importance of graph few-shot learning. (4) FirstMM-DB [28] is the standard 3D point cloud data for link prediction across graphs, which consists of 41 graphs. (5) Tree-of-Life is a new dataset that we constructed based on 1,840 protein interaction networks, each originating from a different species [66]. Since node features are not provided, we use node degrees instead. Details are in Appendix F.

**Experimental setup.** We follow the standard episode training for few-shot learning [42]. We refer the readers to Section 3.1 for task setups given various graph meta learning problems. Note that in order to simulate the real-world few-shot graph learning settings, we do not use the full set of labels provided in the dataset. Instead, we select a partition of it by constructing a fixed number of meta-tasks. Here, we describe the various parameters used in this study. (1) Node classification. For disjoint label setups, we sample 5 labels for meta-testing, 5 for meta-validation, and use the rest for meta-training. In each task, 2-ways 1-shot learning with 5 gradient update steps in meta-training and 10 gradient update steps in meta-testing is used for synthetic datasets. 3-ways 3-shots learning with 10 gradient update steps in meta-training and 20 gradient update steps in meta-testing are used for real-world datasets disjoint label settings. For multiple graph shared labels setups, 10% (10%) of all graphs are held out for testing (validation). The remaining graphs are used for training. For fold-PPI, we use the average of ten 2-way protein function tasks. (2) Link prediction. 10% of graphs are held out for testing and another 10% for validation. For each graph, the support set consists of 30% edges and the query set 70%. Negative edges are sampled randomly to match the same number of positive edges, and we follow the experiment setting from [61] to construct the training graph. We use 16-shots for each task, *i.e.*, using only 32 node pairs to predict links for an unseen graph. 10 gradient update steps in meta-training and 20 gradient update steps in meta-testing are used. Each experiment is repeated five times to calculate the standard deviation of the results. Hyperparameters selection and a recommended set of them are in Appendix G.

**Nine baseline methods.** Meta-Graph [3] injects graph signature in VGAE [21] to do few-shot multi-graph link prediction. Meta-GNN [63] applies MAML [12] to Simple Graph Convolution (SGC) [52]. Few-shot Graph Isomorphism Network (FS-GIN) [54] applies GIN on the entire graph and retrieve the few-shot nodes to propagate loss and enable learning. Similarly, Few-shot SGC (FS-SGC) [52] switches GIN to SGC for GNN encoder. Note that the previous four baselines only work in a

**Table 2: Graph meta-learning performance on synthetic datasets.** Reported is multi-class classification accuracy (five-fold average) for 1-shot node classification in various graph meta-learning problems. N/A means the method does not apply for the problem. Disjoint label problems use a 2-way setup. In the shared label problem, the cycle graph has 17 labels and the BA graph has 10 labels. The results use 5 gradient update steps in meta-training and 10 gradient update steps in meta-testing. Full table with standard deviations is in Appendix K.

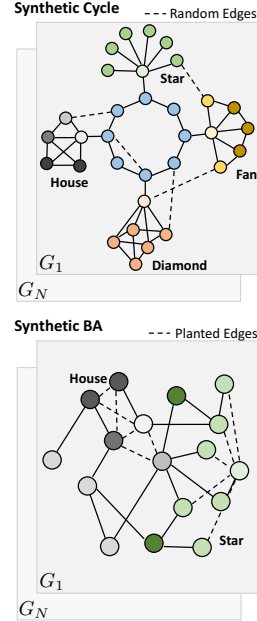

**Figure 3:** Synthetic Cycle and Barabási-Albert datasets.

| Graph Meta-Learning Problem | Single graph Disjoint labels | | Multiple graphs Shared labels | | Multiple graphs Disjoint labels | |
|---|---|---|---|---|---|---|
| Prediction Task | Node | | Node | | Node | |
| Base graph | Cycle | BA | Cycle | BA | Cycle | BA |
| G-META (Ours) | 0.872 | **0.867** | **0.542** | **0.734** | **0.767** | **0.867** |
| Meta-Graph | N/A | N/A | N/A | N/A | N/A | N/A |
| Meta-GNN | 0.720 | 0.694 | N/A | N/A | N/A | N/A |
| FS-GIN | 0.684 | 0.749 | N/A | N/A | N/A | N/A |
| FS-SGC | 0.574 | 0.715 | N/A | N/A | N/A | N/A |
| KNN | **0.918** | 0.804 | 0.343 | 0.710 | 0.753 | 0.769 |
| No-Finetune | 0.509 | 0.567 | 0.059 | 0.265 | 0.592 | 0.577 |
| Finetune | 0.679 | 0.671 | 0.385 | 0.517 | 0.599 | 0.629 |
| ProtoNet | 0.821 | 0.858 | 0.282 | 0.657 | 0.749 | 0.866 |
| MAML | 0.842 | 0.848 | 0.511 | 0.726 | 0.653 | 0.844 |

**Table 3: Graph meta-learning performance on real-world datasets.** Reported is multi-class classification accuracy (five-fold average) and standard deviation. N/A means the method does not work in the graph meta-learning problem. Note that ProtoNet and MAML can be considered as ablation studies of G-META. Further details on performance are in Appendix I.

| Graph Meta-Learning Problem | Single graph Disjoint labels | Multiple graphs Shared labels | Multiple graphs Disjoint labels | Multiple graphs Shared labels | Multiple graphs Shared labels |
|---|---|---|---|---|---|
| Prediction Task | Node | Node | Node | Link | Link |
| Dataset | ogbn-arxiv | Tissue-PPI | Fold-PPI | FirstMM-DB | Tree-of-Life |
| G-META (Ours) | $\mathbf{0.451}_{\pm 0.032}$ | $\mathbf{0.768}_{\pm 0.029}$ | $\mathbf{0.561}_{\pm 0.059}$ | $\mathbf{0.784}_{\pm 0.028}$ | $\mathbf{0.722}_{\pm 0.032}$ |
| Meta-Graph | N/A | N/A | N/A | $0.719_{\pm 0.020}$ | $0.705_{\pm 0.004}$ |
| Meta-GNN | $0.273_{\pm 0.122}$ | N/A | N/A | N/A | N/A |
| FS-GIN | $0.336_{\pm 0.042}$ | N/A | N/A | N/A | N/A |
| FS-SGC | $0.347_{\pm 0.005}$ | N/A | N/A | N/A | N/A |
| KNN | $0.392_{\pm 0.015}$ | $0.619_{\pm 0.025}$ | $0.433_{\pm 0.034}$ | $0.603_{\pm 0.072}$ | $0.649_{\pm 0.012}$ |
| No-Finetune | $0.364_{\pm 0.014}$ | $0.516_{\pm 0.006}$ | $0.376_{\pm 0.017}$ | $0.509_{\pm 0.006}$ | $0.505_{\pm 0.001}$ |
| Finetune | $0.359_{\pm 0.010}$ | $0.521_{\pm 0.013}$ | $0.370_{\pm 0.022}$ | $0.511_{\pm 0.007}$ | $0.504_{\pm 0.003}$ |
| ProtoNet | $0.372_{\pm 0.017}$ | $0.546_{\pm 0.025}$ | $0.382_{\pm 0.031}$ | $0.779_{\pm 0.020}$ | $0.697_{\pm 0.010}$ |
| MAML | $0.389_{\pm 0.021}$ | $0.745_{\pm 0.051}$ | $0.482_{\pm 0.062}$ | $0.758_{\pm 0.025}$ | $0.719_{\pm 0.012}$ |

few graph meta-learning problems. We also test on different meta-learning models, using the top performing ones in [42]. We operate on subgraph level for them since it allows comparison in all problems. No-Finetune performs training on the support set and use the trained model to classify each query example, using only meta-testing set. KNN [10, 42] first trains a GNN using all data in the meta-training set and it is used as an embedding function. Then, it uses the label of the voted K-closest example in the support set for each query example. Finetune [42] uses the embedding function generated from meta-training set and the models are then finetuned on the meta-testing set. ProtoNet [37] applies prototypical learning on each subgraph embeddings, following the standard few-shot learning setups. MAML [12] switches ProtoNet to MAML as the meta-learner. Note that the baselines ProtoNet and MAML can be considered as an ablation of G-META removing MAML and Prototypical loss respectively. All experiments use the same number of query points per label. We use multi-class accuracy metric.

## 6.1 Results

**Overview of results.** Synthetic dataset results are reported in Table 2 and real-world datasets in Table 3. We see that G-META can consistently achieve the best accuracy in almost all tested graph

meta-learning problems and on both node classification and link prediction tasks. Notably, G-META achieves a 15.1% relative increase in the single graph disjoint setting, a 16.3% relative increase in the multiple graph disjoint label setting, over the best performing baseline. This performance boost suggests that G-META can learn across graphs and disjoint labels. Besides, this increase also shows G-META obtains good predictive performance, using only a few gradient update steps given a few examples on the target tasks.

**G-META captures local structural information.** In the synthetic datasets, we see that Meta-GNN, FS-GIN, and FS-SGC, which base on the entire graph, are inferior to subgraph-based methods, such as G-META. This finding demonstrates that subgraph embedding can capture the local structural roles, whereas using the entire graph cannot. It further supports our theoretical motivation. In the single graph disjoint label setting, KNN achieves the best result. This result suggests that the subgraph representation learned from the trained embedding function is sufficient to capture the structural roles, further confirming the usefulness of local subgraph construction.

**G-META is highly predictive, general graph meta-learning method.** Across meta-learning models, we observe G-META is consistently better than others such as MAML and ProtoNet. MAML and ProtoNet have volatile results across different problems and tasks, whereas G-META is stable. This result confirms our analysis of using the prototypical loss to leverage the label inductive bias and MAML to transfer knowledge across graphs and labels. By comparing G-META to two relevant existing works Meta-GNN and Meta-Graph, we see G-META can work across different problems. In contrast, the previous two methods are restricted by the meta graph learning problems. In real-world datasets, we observe No-Finetune is better than Finetune in many problems (ogbn-arxiv & Fold-PPI). This observation shows that the meta-training datasets bias the meta-testing result, suggesting the importance of meta-learning algorithm to achieve positive transfer in meta graph learning.

**Local subgraphs are vital for successful graph few-shot learning.** We observe that Meta-GNN, FS-GIN, and FS-SGC, which operate on the entire graph, perform poorly. In contrast, G-META can learn transferable signal, even when label sets are disjoint. Since the problem requires learning using only 3-shots in a large graph of around 160 thousands nodes and 1 million edges (ogbn-arxiv), we posit that operating GNN on the entire graph would not propagate useful information. Hence, the performance increase suggests that local subgraph construction is useful, especially in large graphs. Finally, in the Tree-of-Life link prediction problem, we studied the largest-ever graph meta-learning dataset, spanning 1,840 PPI graphs. This experiment also supports the scalability of G-META, which is enabled by the local subgraphs.

**Ablation and parameter studies.** We conduct ablations on two important components, optimization-based meta-learning and prototypical loss. Note that baseline ProtoNet and MAML are the ablations of G-META. We find that both components are important to improve predictive performance. To study parameters, we first vary the size of subgraph size $h$. We find that $h = 2$ has a better performance across datasets than $h = 1$. When $h = 3$, in some dataset, it introduces noise and decreases the performance while in other one, it can improve performance since it includes more information. We suggest to use $h = 2$ since it has the most stable performance. We then vary the number of shots $k$, and observes a linear trend between $k$ and the predictive performance. The detailed quantitative results on the parameter studies are provided in the Appendix J.

## 7   Conclusion

We introduce G-META, a scalable and inductive meta learning method for graphs. Unlike earlier methods, G-META excels at difficult few-shot learning tasks and can tackle a variety of meta learning problems. The core principle in G-META is to use local subgraphs to identify and transfer useful information across tasks. The local subgraph approach is fundamentally different from prior studies, which use entire graphs and capture global graph structure at the loss of finer topological shapes. Local subgraphs are theoretically justified and stem from our observation that evidence for a prediction can be found in the local subgraph surrounding the target entity. G-META outperforms nine baselines across seven datasets, including a new and original dataset of 1,840 graphs.

## Broader Impact

Graphs represent an incredibly powerful data representation that has proved useful for numerous domains and application areas. At the same time, meta learning is a key area of machine learning research that has already demonstrated great potential for problems, such as few-shot image recognition and neural architecture search.

**G-META advances ML research.** Our present work is at an exciting yet underexplored intersection of graph ML and meta learning, advancing the state-of-the-art and enabling practical applications of meta learning for large graph datasets. State-of-the-art GNN methods do not work when there are only a handful of labels available. Our work fills in this gap by providing a novel approach to leverage related information such as related graphs or other labels sets to aid the graph few-shot learning. Numerous real-world graphs are only associated with scarce labels of nodes or have a large percentage of missing links. Low-resource constraint (*e.g.*, the sparsity of graphs/scarcity of labels) is a fundamental challenge in real-world graph applications. The goal of graph meta-learning is to solve key graph ML tasks, such as node classification and link prediction, under these constraints.

We envision numerous impactful applications of our work, as already demonstrated in the paper. We provide a brief overview of key applications for graph meta-learning, and G-META in particular.

**G-META can expedite scientific discovery.** Science is filled with complex systems that are modeled by graphs. The labels are usually obtained through resource-intensive experiments in laboratories. Many experiments, such as those to measure protein-protein interactions (studied in the paper), cannot yet be automated. Because of that, labels (*i.e.*, a variety of molecular properties that need to be discovered for downstream problems like drug discovery [44]) are expensive to obtain and thus are very scarce. G-META can accelerate scientific discovery by learning from related sources and quickly adapting to a new, never-before-seen task of interest given only a handful of examples. For example, G-META can better annotate rare disease pathways by transferring knowledge from the mouse PPI network to the human PPI network [33]. Further, PPI networks of many species are highly incomplete, even for humans, less than 30% of all pairwise combinations of proteins have been tested for interaction so far [25]. Graph meta-learning can help label sparse networks by learning from a handful of other networks that are well-annotated.

**G-META can bring economic values.** While graphs have helped businesses make better products (*e.g.*, [56, 26]), many graph-structured data resources remain under-utilized because of the sparsity and scarcity of labels. G-META can help tackle this problem. For example, it can improve recommendation for a new set of products that only have a few user behavior records by transferring knowledge from previous product sets in the user products recommendation networks, or help a business expand to new locations by learning from the interconnected data about the current location.

**G-META can improve equality.** G-META can be used to facilitate the development of world regions by quickly learning from the developed regions. While this has previously been successfully demonstrated on satellite image data [20], we see many new opportunities for graph data, such as identifying what transportation lines (links) should be prioritized to add in a rural region by learning from infrastructure networks of developed regions that were similarly rural before.

Finally, we briefly comment on potential risks of having a powerful graph meta-learning predictor. (1) Negative transfer: Meta-learning models are tricky to train and may vary across datasets and domains. It can lead to negative transfer. (2) Misuse of the methodology: Meta-learning is not universal. It works in specific settings. For example, G-META is applied to few-shot settings. When labels are abundant, it might not yield considerable benefits. (3) Adversarial attacks: There are no studies on how adversarial attacks would affect meta-learning algorithms and graph algorithms in particular. We posit that in the few-shot setting, it may be particularly easy to attack the graph since each labeled example is vital for prediction.

## Acknowledgments and Disclosure of Funding

This work is supported, in part, by NSF grant nos. IIS-2030459 and IIS-2033384, and by the Harvard Data Science Initiative. The content is solely the responsibility of the authors. We thank Chengtai Cao, Avishek Joey Bose, Xiang Zhang for helpful discussions.

## Footnotes

[1]Code and datasets are available at https://github.com/mims-harvard/G-Meta.

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
