[Supplementary Material]

# Appendix A  Further Details on Notation

**Table 4: The notation used in the paper.**

| Notation | Description |
|---|---|
| $\mathcal{G}, G$ | A set of graphs, graph $G \in \mathcal{G}$ |
| $\mathcal{V}, \mathcal{E}, \mathbf{X}$ | A set of nodes, edges, and feature matrix for graph $G$ |
| $\mathcal{Y}, \mathrm{Y}, Y$ | An entire set of labels, a label set for a task, and label $Y \in \mathrm{Y}$ |
| $\mathcal{S}, S$ | A set of subgraphs, subgraph $S \in \mathcal{S}$ |
| $u, v$ | Nodes $u$ and $v$ |
| $i, j$ | Prediction tasks $i$ and $j$ |
| $f_\theta$ | GNN model $f$ parameterized by $\theta$ |
| $\mathcal{D}, \mathcal{D}_{\text{train}}, \mathcal{D}_{\text{val}}, \mathcal{D}_{\text{test}}$ | An entire meta-set, meta-training set, meta-validation set, and meta-test set |
| $\mathcal{T}_i, \mathcal{T}_i^{\text{support}}, \mathcal{T}_i^{\text{query}}$ | Task $i$, support set, and query set for task $i$ |
| $N, M$ | Number of graphs and labels |
| $\mathrm{N}, \mathrm{K}_{\text{support}}, \mathrm{K}_{\text{query}}$ | N-way, $\mathrm{K}_{\text{support}}$-shot prediction for support set and $\mathrm{K}_{\text{query}}$-shot for query set |
| $\eta$ | Number of steps in the inner-loop (see Algorithm 1) |
| $\mathcal{L}_{\text{support}}, \mathcal{L}_{\text{query}}$ | Support loss and query-set loss |
| $\mathbf{H}$ | A matrix of centroid embeddings |
| $\mathbf{C}$ | A matrix of prototype embeddings |
| $\mathbf{p}$ | A class probability distribution vector |
| $\mathbf{y}_{\text{support}}, \mathbf{y}_{\text{query}}$ | Support and query set label vector |
| $\alpha, \beta$ | Learning rate of inner and outer loop update (see Algorithm 1) |

# Appendix B  Further Details on the Applicability of Methods for Meta Learning on Graphs

**Table 5: A comparison of notable existing methods in the context of different graph meta-learning problems.** We see G-META is able to tackle all problems whereas existing methods cannot.

| Graph Meta-Learning Problem | Single graph Shared labels | Single graph Disjoint labels | Multiple graphs Shared labels | Multiple graphs Disjoint labels |
|---|---|---|---|---|
| GNN, *e.g.*, [22, 54, 52] | ✔ | ✗ | ✗ | ✗ |
| Meta-GNN [63] | ✗ | ✔ | ✗ | ✗ |
| Meta-Graph [3] | ✗ | ✗ | ✔ | ✗ |
| G-META (Ours) | ✔ | ✔ | ✔ | ✔ |

# Appendix C  Assumptions and Proof of Theorem 1

## C.1  Assumptions

We use a popular GCN model [22] as the exemplar GNN model. The $l$-th layer GCN propagation rule is defined as: $\mathbf{H}^{(l+1)} = \sigma(\hat{\mathbf{A}} \mathbf{H}^{(l)} \mathbf{W}^{(l)})$, where $\mathbf{H}^{(l)}, \mathbf{W}^{(l)}$ are node embedding and parameter weight matrices at layer $l$, respectively, and $\hat{\mathbf{A}} = \mathbf{D}^{-1}\mathbf{A}$ is the normalized adjacency matrix. Following [48], throughout these derivations, we assume that $\sigma$ is an identity function and that $\mathbf{W}$ an identity matrix.

## C.2  Definitions and Notation

**Definition (Node Influence)**: *Node influence $I_{u,v}$ of $v$ on $u$ in the final GNN output is: $I_{u,v} = \|\partial \mathbf{x}_u^{(\infty)}/\partial \mathbf{x}_v^{(\infty)}\|$, where the norm is any subordinate norm and the Jacobian measures how a change in $v$ translates to a change in $u$ [48].*

**Definition (Graph Influence)**: *Graph influence $I_G$ on $u$ is: $I_G(u) = \|[I_{u,v_1}, \ldots, I_{u,v_n}]\|_1$, where $[I_{u,v_1}, \ldots, I_{u,v_n}]$ is a vector representing the influence of other nodes on $u$.*

The graph influence of graph $G$ on node $u$ is: $I_G(u) = \sum_{i \in \mathcal{V}} I_{u,v_i}$. Similarly, the influence of a $h$-hop neighborhood subgraph $S_u$ on node $u$ is: $I_{S_u}(u) = \sum_{i \in \mathcal{V}^u} I_{u,v_i}$ where $\mathcal{V}^u$ contains nodes that are at most $h$ hops away from node $u$, *i.e.*, $\{i_x | d(i_x, u) \leq h\}$.

**Definition (Graph Influence Loss)**: *Graph influence loss $R_h$ is defined as: $R_h(u) = I_G(u) - I_{S_u}(u)$, where $I_G(u)$ is the influence of entire graph $G$ and $I_{S_u}(u)$ is the influence of subgraph $S_u$.*

Notationally, we denote $m$ paths between node $u$ and $v$ as: $p^1, \ldots, p^m$. We use $p_i^x, p_j^x, \ldots, p_{n_v}^x$ to represent nodes occurring on path $p^x$ and use $n_v$ to denote the length of the path.

## C.3 Theorem and Proof

**Theorem 1 (Decaying Property of Node Influence).** *Let $t$ be a path between node $u$ and node $v$ and let $D_{\mathrm{GM}}^t$ be a geometric mean of node degrees occurring on path $t$. Let $D_{\mathrm{GM}}^{t_*} = \min_t \{D_{\mathrm{GM}}^t\}$ and $h_* = d(u, v)$. Consider the node influence $I_{u,v}$ from $v$ to $u$. Then, $I_{u,v} \leq C/(D_{\mathrm{GM}}^{t_*})^{h_*}$.*

*Proof.* Using the GCN propagation rule, we see that node $u$'s output is defined as:

$$\mathbf{x}_u^{(\infty)} = \frac{1}{D_{uu}} \sum_{i \in \mathcal{N}(u)} a_{ui} \mathbf{x}_i^{(\infty)},$$

where $a_{ui}$ is the edge weight between node $u$ and node $i$. In our datasets and many other real-world graphs, all edges have the same weight 1.

By an expansion of nodes in the neighbor $\mathcal{N}(u)$, we have:

$$\mathbf{x}_u^{(\infty)} = \frac{1}{D_{uu}} \sum_{i \in \mathcal{N}(u)} a_{ui} \frac{1}{D_{ii}} \sum_{j \in \mathcal{N}(i)} a_{ij} \mathbf{x}_j^{(\infty)}.$$

In the same logic, it can be further expanded as:

$$\mathbf{x}_u^{(\infty)} = \frac{1}{D_{uu}} \sum_{i \in \mathcal{N}(u)} a_{ui} \frac{1}{D_{ii}} \sum_{j \in \mathcal{N}(i)} a_{ij} \cdots \frac{1}{D_{mm}} \sum_{o \in \mathcal{N}(m)} a_{mo} \mathbf{x}_o^{(\infty)}. \tag{1}$$

The node influence $I_{u,v} = \|\frac{\partial \mathbf{x}_u^{(\infty)}}{\partial \mathbf{x}_v^{(\infty)}}\|$ can then be computed via:

$$
\begin{aligned}
\|\frac{\partial \mathbf{x}_u^{(\infty)}}{\partial \mathbf{x}_v^{(\infty)}}\| &= \|\frac{\partial}{\partial \mathbf{x}_v^{(\infty)}} \left( \frac{1}{D_{uu}} \sum_{i \in \mathcal{N}(u)} a_{ui} \frac{1}{D_{ii}} \sum_{j \in \mathcal{N}(i)} a_{ij} \cdots \frac{1}{D_{mm}} \sum_{o \in \mathcal{N}(m)} a_{mo} \mathbf{x}_o^{(\infty)} \right)\| \quad [1]\\
&= \|\frac{\partial}{\partial \mathbf{x}_v^{(\infty)}} \left( \left( \frac{1}{D_{uu}} a_{(up_i^1)} \frac{1}{D_{p_i^1 p_i^1}} a_{(p_i^1 p_j^1)} \cdots \frac{1}{D_{p_{n_1}^1 p_{n_1}^1}} a_{(p_{n_1}^1 v)} \mathbf{x}_v^{(\infty)} \right) \right.\\
&\quad + \cdots + \left. \left( \frac{1}{D_{uu}} a_{(up_i^m)} \frac{1}{D_{p_i^m p_i^m}} a_{(p_i^m p_j^m)} \cdots \frac{1}{D_{p_{n_m}^m p_{n_m}^m}} a_{(p_{n_m}^m v)} \mathbf{x}_v^{(\infty)} \right) \right)\| \quad [2]\\
&= \|\frac{\partial \mathbf{x}_v^{(\infty)}}{\partial \mathbf{x}_v^{(\infty)}}\| \cdot | \left( \frac{1}{D_{uu}} a_{(up_i^1)} \frac{1}{D_{p_i^1 p_j^1}} a_{(p_i^1 p_j^1)} \cdots \frac{1}{D_{p_{n_1}^1 p_{n_1}^1}} a_{(p_{n_1}^1 v)} \right)\\
&\quad + \cdots + \left( \frac{1}{D_{uu}} a_{(up_i^m)} \frac{1}{D_{p_i^m p_i^m}} a_{(p_i^m p_j^m)} \cdots \frac{1}{D_{p_{n_m}^m p_{n_m}^m}} a_{(p_{n_m}^m v)} \right) | \quad [3]\\
&= | \left( \frac{1}{D_{uu} D_{p_i^1 p_i^1} \cdots D_{p_{n_1}^1 p_{n_1}^1}} a_{(up_i^1)} a_{(p_i^1 p_j^1)} \cdots a_{(p_{n_1}^1 v)} \right)
\end{aligned}
$$

$$+\cdots+\left(\frac{1}{D_{uu}D_{p_i^1 p_j^1}\cdots D_{p_{n_m}^m p_{n_m}^m}}a_{(up_i^m)}a_{(p_i^m p_j^m)}\cdots a_{(p_{n_m}^m v)}\right)| \tag{4}$$

$$\leq |m*\max\left(\left(\frac{1}{D_{uu}D_{p_i^1 p_i^1}\cdots D_{p_{n_1}^1 p_{n_1}^1}}a_{(up_i^1)}a_{(p_i^1 p_j^1)}\cdots a_{(p_{n_1}^1 v)}\right)\right.$$

$$\left.,\cdots,\left(\frac{1}{D_{uu}D_{p_i^1 p_i^1}\cdots D_{p_{n_m}^m p_{n_m}^m}}a_{(up_i^m)}a_{(p_i^m p_j^m)}\cdots a_{(p_{n_m}^m v)}\right)\right)| \tag{5}$$

$$= |m*\left(\frac{1}{D_{uu}D_{p_i^{t*} p_i^{t*}}\cdots D_{p_{n_*}^{t*} p_{n_*}^{t*}}}a_{(up_i^{t*})}a_{(p_i^{t*} p_j^{t*})}\cdots a_{(p_{n_m}^{t*} v)}\right)| \tag{6}$$

$$= |C*\left(\frac{1}{\sqrt[n_*]{D_{uu}D_{p_i^{t*} p_i^{t*}}\cdots D_{p_{n_*}^{t*} p_{n_*}^{t*}}}}\right)^{n_*}| \tag{7}$$

$$= C/(D_{\mathrm{GM}}^{t*})^{n_*} \tag{8}$$

$$\leq C/(D_{\mathrm{GM}}^{t*})^{h_*}. \tag{9}$$

$\square$

**Explanations of the proof.** (Line 1) Substitution of the term $\mathbf{x}_u^{(\infty)}$ via Equation 1. (Line 2) Since we are calculating the gradient between two feature vectors $\mathbf{x}_v^{(\infty)}$ and $\mathbf{x}_u^{(\infty)}$, the partial derivative on nodes that are not in the paths $p^1,...,p^m$ between node $u$ and $v$ becomes 0 and these nodes are thus removed. Note that each term in line 2 is for one path between node $u$ and $v$. This also means that the feature influence can be decomposed into the sum of all paths feature influence. (Line 3) We separate the scalar terms and the derivative term within the matrix norm and then uses the absolute homogeneous property (*i.e.*$\|\alpha\mathbf{A}\| = |\alpha|\|\mathbf{A}\|$) of the matrix norm to convert them into the matrix norm of Jacobian matrix times the absolute value of the scalar. (Line 4) We know the Jacobian of the same vectors $\frac{\partial \mathbf{x}_v^{(\infty)}}{\partial \mathbf{x}_v^{(\infty)}}$ is the identity matrix $\mathbf{I}$ and we know for any subordinate norm ($\|\mathbf{A}\| = \sup_{\|\mathbf{x}\|=1}\{\|\mathbf{A}\mathbf{x}\|\}$), we have $\|\mathbf{I}\| = 1$. Therefore, $\|\frac{\partial \mathbf{x}_v^{(\infty)}}{\partial \mathbf{x}_v^{(\infty)}}\| = 1$. We also move the degree and edge weight terms around to group each together for each path. (Line 5) Notice now the equation becomes the sum of terms around paths between node $u$ and $v$. We can identify the maximum of these terms and it is smaller than the $m$ times the maximum term. We denote the term with the maximum value as the path $p^{t*}$. (Line 6) Clean the notations. (Line 7) We move all the constant as $C$ in the last line. Note that in all of our datasets and many real-world datasets, the network is usually binary and the edge weight $a$ are non-negative (in many cases $a = 1$), thus we remove the absolute values. (Line 8) We rephrase the degree term into geometric mean format $1/D_{GM}$. (Line 9) $h_*$ is the shortest path and shorter than $n_*$, thus the inequality. We then obtain the bound of the node influence.

Note that if we assume degree of nodes along paths are random, then the $p^{t*}$, which is the path that has the smallest geometric mean of node degrees, is the shortest path from node $u$ to $v$.

## Appendix D  Theorem 2 and its Proof

**Theorem 2 (Local Subgraph Preservation Property).** *Let $S_u$ be a local subgraph for node $u$ with neighborhood size $h$. Let node $v$ be defined as: $v = \operatorname{argmax}_w(\{I_{u,w}|w \in \mathcal{V} \setminus \mathcal{V}^u\})$. Let $\bar{t}$ be a path between $u$ and $v$ and let $D_{\mathrm{GM}}^{\bar{t}}$ be a geometric mean of node degrees occurring on path $\bar{t}$. Let $D_{\mathrm{GM}}^{\bar{t}*} = \min_{\bar{t}}\{D_{\mathrm{GM}}^{\bar{t}}\}$. The following holds: $R_h(u) \leq C/(D_{\mathrm{GM}}^{\bar{t}*})^{h+1}$.*

*Proof.*

$$R_h(u) = I_G(u) - I_{S_u}(u) \tag{1}$$

$$= \left(\|\frac{\partial \mathbf{x}_u^{(\infty)}}{\partial \mathbf{x}_1^{(\infty)}}\| + \cdots + \|\frac{\partial \mathbf{x}_u^{(\infty)}}{\partial \mathbf{x}_n^{(\infty)}}\|\right) - \left(\|\frac{\partial \mathbf{x}_u^{(\infty)}}{\partial \mathbf{x}_{i_1}^{(\infty)}}\| + \cdots + \|\frac{\partial \mathbf{x}_u^{(\infty)}}{\partial \mathbf{x}_{i_m}^{(\infty)}}\|\right) \tag{2}$$

$$= \|\frac{\partial \mathbf{x}_u}{\partial \mathbf{x}_{t_1}}\| + \|\frac{\partial \mathbf{x}_u}{\partial \mathbf{x}_{t_2}}\| + \cdots + \|\frac{\partial \mathbf{x}_u}{\partial \mathbf{x}_{t_{n-m}}}\| \qquad [3]$$

$$\leq C_{t_1}/(D_{\mathrm{GM}}^{t_1})^{h_{t_1}} + \cdots + C_{t_{n-m}}/(D_{\mathrm{GM}}^{t_{n-m}})^{h_{t_{n-m}}} \qquad [4]$$

$$\leq (n-m) * C_{\bar{t}_*}/(D_{\mathrm{GM}}^{\bar{t}_*})^{h_{\bar{t}_*}} \qquad [5]$$

$$\leq (n-m) * C_{\bar{t}_*}/(D_{\mathrm{GM}}^{\bar{t}_*})^{h+1} \qquad [6]$$

$$= C/(D_{\mathrm{GM}}^{\bar{t}_*})^{h+1} \qquad [7]$$

$\square$

**Explanation of the proof.** (Line 1) Definition of graph influence loss. (Line 2) Substitution of definition. (Line 3) Subtract same node influence term from the subgraph and the entire network. (Line 4) Use Theorem 1. (Line 5) They are smaller than $(n-m)$ times the maximum term, which is the node that has the highest node influence in the node set that is outside of the immediate neighborhood of $u$. (Line 6) We know nodes in the outside of the immediate neighborhood of $u$ is more than $h$ hops away from the node $u$, hence the path length $h_{t_*}$ between the maximum influence node to the node $u$ is larger than $h$. (Line 7) Move the constant term and we have the result.

## Appendix E    Algorithm Overview

We provide the pseudo-code for G-META in Algorithm 1.

---

**Algorithm 1:** G-META Algorithm. Steps in the algorithm correspond to steps in Figure 2.

---

**Input**: Graphs $\mathcal{G} = \{G_1, ..., G_N\}$; Randomly initialized $\theta : [\theta_{\mathrm{GNN}}]$.
$S_1, S_2, ..., S_n = \mathrm{Subgraph}(\mathcal{G})$ via step #1          // Local subgraph construction
$\{\mathcal{T}\} = \{\mathcal{T}_1, \mathcal{T}_2, ..., \mathcal{T}_m\} \sim p(\mathcal{T})$ via step #1          // Meta-task construction
**while** not done **do**
    $\{\mathcal{T}_s\} \leftarrow \mathrm{sample}(\{\mathcal{T}\})$          // Sample a batch of tasks
    **for** $\mathcal{T}_i \in \{\mathcal{T}_s\}$ **do**
        $(\{S\}_{\mathrm{support}}, \mathbf{y}_{\mathrm{support}}) \leftarrow \mathcal{T}_i^{\mathrm{support}}$          // Mini-batching support subgraphs
        $(\{S\}_{\mathrm{query}}, \mathbf{y}_{\mathrm{query}}) \leftarrow \mathcal{T}_i^{\mathrm{query}}$          // Mini-batching query subgraphs
        $\theta_0 = \theta$
        **for** j in 1, ..., $\eta$ **do**          // Update step $j$
            $\mathbf{H}_{\mathrm{support}} \leftarrow \mathrm{GNN}_{\theta_{j-1}}(\{S\}_{\mathrm{support}})_{\mathrm{centroid}}$ via step #2 and #3
            $\mathbf{C} = \frac{1}{N_{k_{\mathrm{support}}}} \sum(\mathbf{H}_{\mathrm{support}})$ via step #4          // Support prototypes
            $\mathbf{p} = \frac{\exp(-\|\mathbf{H}_{\mathrm{support}} - \mathbf{C}\|)}{\sum_{\mathbf{Y}_i} \exp(-\|\mathbf{H}_{\mathrm{support}} - \mathbf{C}\|)}$ via step #5
            $\mathcal{L}_{\mathrm{support}} = \mathrm{L}(\mathbf{p}, \mathbf{y}_{\mathrm{support}})$ via step #5
            $\theta_j = \theta_{j-1} - \alpha \nabla \mathcal{L}_{\mathrm{support}}$ via step #6          // Inner loop update
            $\mathbf{H}_{\mathrm{query}} \leftarrow \mathrm{GNN}_{\theta_j}(\{S\}_{\mathrm{query}})_{\mathrm{centroid}}$ via step #7 and #8
            $\mathbf{p} = \frac{\exp(-\|\mathbf{H}_{\mathrm{query}} - \mathbf{C}\|)}{\sum_{\mathbf{Y}_i} \exp(-\|\mathbf{H}_{\mathrm{query}} - \mathbf{C}\|)}$ via step #9
            $\mathcal{L}_{\mathrm{query}}^{ij} \leftarrow \mathrm{L}(\mathbf{p}, \mathbf{y}_{\mathrm{query}})$ via step #9
        **end**
    **end**
    $\theta = \theta - \beta \nabla \sum_i \mathcal{L}_{\mathrm{query}}^{iu}$ via step #10          // Outer loop update
**end**

---

## Appendix F    Further Details on Datasets

We proceed by describing the construction and processing of synthetic as well as real-world datasets. G-META implementation as well as all datasets and the relevant data loaders are available at https://github.com/mims-harvard/G-Meta.

### F.1 Synthetic Datasets

We have two synthetic datasets where labels are depended on local structural roles. They are to show G-META's ability to capture local network structures. For the first synthetic dataset, we use the graphs with planted structural equivalence from GraphWave [9]. We use a cycle basis network and attach a distribution of shapes: House, Star, Diamond, Fan on the cycle basis. The label of each node is the structural role in different shapes. Hence, the label reflects the local structural information. We also add n random edges to add noise. For the single graph and disjoint label setting, we use 500 nodes for the cycle basis, and add 100 shapes for each type with 1,000 random edges. In the multiple graph setting, we sample 10 graphs with varying distribution of number of shapes for each graph. Each graph uses 50 nodes for the cycle basis, and add randomly generated [1-15] shapes for each type with 100 random edges. There are 17 labels. To model local structural information under a more realistic homophily network, we first construct a Barabási-Albert (BA) network with 200 nodes and 3 nodes are preferential attached based on the degrees. Then, we plant shapes to the BA network by first sampling nodes and adding edges corresponding to the shapes. Then, to generate label, we compute the Graphlet Distribution Vector [31] for each node, which characterizes the local network structures and then we apply spectral clustering on this vector to generate the labels. There are 10 labels in total. For multiple graph setting, the same varying distribution of numbers of shapes as in the cycle dataset are used to plant each BA network. See Figure 3 for a visual illustration of synthetic datasets.

### F.2 Real-world Datasets and Novel Graph Meta-Learning Datasets

We use three real world datasets for node classification and two real world datasets for link prediction to evaluate G-META. (1) arXiv is a citation network from the entire Computer Science arXiv papers, where features are title and abstract word embeddings, and labels are the subject areas [17]. (2) Tissue-PPI is 24 protein-protein interaction networks from different tissues, where features are gene signatures and labels are gene ontology functions [67, 15]. Each label is a binary protein function classification task. We select the top 10 balanced tasks. (3) Fold-PPI is a novel dataset, which we constructed for the multiple graph and disjoint label setting. It has 144 tissue networks [67], and the labels are classified using protein structures defined in SCOP database [2]. We screen fold groups that have more than 9 unique proteins across the networks. It results in 29 unique labels. The features are conjoint triad protein descriptor [36]. In Fold-PPI, the majority of the nodes do not have associated labels. Note that G-META operates on label scarce settings. (4) For link prediction, the first dataset FirstMM-DB [28] is the standard 3D point cloud data, which consists of 41 graphs. (5) The second link prediction dataset is the Tree-of-Life dataset. This is a new dataset, which we constructed based on 1,840 protein interaction networks (PPIs), each originating from a different species [66]. Node features are not provided, we use node degrees instead.

## Appendix G  Further Details on Hyperparameter Selection

We use random hyperparameter search over the following set of hyperparameters. For task numbers in each batch, we use 4, 8, 16, 32, 64; for inner update learning rate, $1 \times 10^{-2}$, $5 \times 10^{-3}$, $1 \times 10^{-3}$, $5 \times 10^{-4}$; for outer update learning rate, $1 \times 10^{-2}$, $5 \times 10^{-3}$, $1 \times 10^{-3}$, $5 \times 10^{-4}$; for hidden dimension, we select from 64, 128, and 256.

For arxiv-ogbn dataset, we set task numbers to 32, inner update learning rate to $1 \times 10^{-2}$, outer update learning rate to $1 \times 10^{-3}$, and hidden dimensionality to 256. For Tissue-PPI dataset, we set task numbers to 4, inner update learning rate to $1 \times 10^{-2}$, outer update learning rate to $5 \times 10^{-3}$, and hidden dimensionality to 128. For Fold-PPI dataset, we set task numbers to 16, inner update learning rate to $5 \times 10^{-3}$, outer update learning rate to $1 \times 10^{-3}$, and hidden dimensionality to 128. For FirstMM-DB dataset, we set task numbers to 8, inner update learning rate to $1 \times 10^{-2}$, outer update learning rate to $5 \times 10^{-4}$, and hidden dimensionality to 128. For Tree-of-Life dataset, we set task numbers to 8, inner update learning rate to $5 \times 10^{-3}$, outer update learning rate to $5 \times 10^{-4}$ and hidden dimensionality to 256.

## Appendix H    Further Details on Baselines

We use nine baselines. (1) Meta-Graph [3] uses VGAE to do few-shot multi-graph link prediction. It uses a graph signature function to capture the characteristics of a graph, which enables knowledge transfer. Then, it applies MAML to learn across graphs. (2) Meta-GNN [63] applies MAML [12] to Simple Graph Convolution(SGC) [52] on the single graph disjoint label problem. (3) Few-shot Graph Isomorphism Network (FS-GIN) [54] applies GIN on the entire graph and retrieve the few-shot nodes to propagate loss and enable learning. Similarly, (4) Few-shot SGC (FS-SGC) [52] switches GIN to SGC for GNN encoder. Note that the previous four baselines only work in a few graph meta learning problems. We also test on different meta-learning models, using the top performing ones in [42]. We operate on subgraph level for them since it allows comparison in all graph meta-learning problems. No-Finetune performs training on the support set and use the trained model to classify each query example, using only meta-testing set, *i.e.* without access to the external graphs or labels to transfer. KNN [10, 42] first trains a GNN using all data in the meta-training set and it is used as an embedding function. Then, during meta-testing, it uses the label of the voted K-closest example in the support set for each query example. Finetune [42] uses the embedding function generated from meta-training set and the models are then finetuned on the meta-testing set. ProtoNet [37] applies prototypical learning on each subgraph embeddings, following the standard few-shot learning setups. MAML [12] switches ProtoNet to MAML model as the meta-learner.

## Appendix I    Further Details on Performance Evaluation

All of our experiments are done on an Intel Xeon CPU 2.50GHz and using an NVIDIA K80 GPU.

For synthetic datasets disjoint label problems, we use a 2-way setup. In the shared label problem, the cycle graph has 17 labels and the BA graph has 10 labels. The results use 5 gradient update steps in meta-training and 10 gradient update steps in meta-testing. For real-world datasets node classification uses 3-shots and link prediction uses 16-shots. For disjoint labels problems, we set it to 3-way classification task. The results use 20 gradient update steps in meta-training and 10 gradient update steps in meta-testing. For Tissue-PPI, we use the average of ten 2-way protein function tasks where each task is performed three times. For link prediction problem, to ensure the support and query set are distinct in all meta-training tasks, we separate a fixed 30% of edges for support set and 70% of edges for query set as a preprocessing step for every graph.

## Appendix J    Parameter Studies

We select Fold-PPI from node classification and FirstMM-DB from link prediction to conduct the parameter studies. We then conduct five runs on each setup and report the average performance below. For varying the number of $k$, we experiment on $k = 1, 3, 10$ for Fold-PPI and $k = 16, 32, 64$ for FirstMM-DB. We find linear trend between $k$ and predictive performance: for Fold-PPI, the performance increases from 0.403 to 0.561 to 0.663; for FirstMM-DB, G-META increases from 0.758 to 0.784 to 0.795. Then, we vary the number of $h$-hops neighbor. We try $h = 1, 2, 3$. For Fold-PPI, the performance is 0.399, 0.561 and 0.427. For FirstMM-DB, the result is 0.616, 0.784 and 0.837. It seems that $h = 1$ has the worst performance but for $h = 3$, it depends on the dataset. For Fold-PPI, $h = 2$ is significantly better than $h = 3$ but it is not the case for FirstMM-DB. Since $h = 2$ has pretty close result with $h = 3$ in FirstMM-DB, we suggest to set $h = 2$ for stable performance.

## Appendix K    Further Results on Synthetic Datasets

Full version is reported in Table 6. The large standard deviation is because we sample only two-labels for meta-testing in one data fold, due to the limit number of labels in synthetic datasets. If the shape structural roles corresponding to the label set in the meta-testing set is structurally distinct from all the meta-training label sets, the performance would be bad since there is no transferability for meta-learner to learn. In our data splits, we find there is one fold that performs bad across all methods, thus the reason for the large standard deviation. For real-world datasets, as the label set is large, we sample more labels for meta-testing (*e.g.*, five labels for 3-way classification where each meta-testing task is of $\binom{5}{3}$ label sets) such that the result is averaged across different label sets. This makes the standard deviation smaller. In both settings, the mean accuracy reflects the predictive performance.

**Table 6: Further results on graph meta-learning performance for synthetic datasets.** Five-fold average multi-class classification accuracy on synthetic datasets for 1-shot node classification in various graph meta-learning problem. N/A means the method does not work in this task setting. The disjoint label setting uses 2-way setup and in the shared labels settings, cycle basis has 17 labels and BA has 10 labels. The results use 5-gradient update steps in meta-training and 10-gradient update steps in meta-testing. See Figure 3 for a visual illustration of synthetic datasets.

| Graph Meta-Learning Problem | Single graph Disjoint labels | | Multiple graphs Shared labels | | Multiple graphs Disjoint labels | |
|---|---|---|---|---|---|---|
| Prediction Task | Node | | Node | | Node | |
| Base graph | Cycle | BA | Cycle | BA | Cycle | BA |
| G-META (Ours) | $0.872_{\pm0.129}$ | $\mathbf{0.867}_{\pm0.147}$ | $\mathbf{0.542}_{\pm0.045}$ | $\mathbf{0.734}_{\pm0.038}$ | $\mathbf{0.767}_{\pm0.178}$ | $\mathbf{0.867}_{\pm0.209}$ |
| Meta-Graph | N/A | N/A | N/A | N/A | N/A | N/A |
| Meta-GNN | $0.720_{\pm0.218}$ | $0.694_{\pm0.112}$ | N/A | N/A | N/A | N/A |
| FS-GIN | $0.684_{\pm0.144}$ | $0.749_{\pm0.106}$ | N/A | N/A | N/A | N/A |
| FS-SGC | $0.574_{\pm0.092}$ | $0.715_{\pm0.100}$ | N/A | N/A | N/A | N/A |
| KNN | $\mathbf{0.918}_{\pm0.063}$ | $0.804_{\pm0.193}$ | $0.343_{\pm0.038}$ | $0.710_{\pm0.027}$ | $0.753_{\pm0.150}$ | $0.769_{\pm0.199}$ |
| No-Finetune | $0.509_{\pm0.011}$ | $0.567_{\pm0.105}$ | $0.059_{\pm0.001}$ | $0.265_{\pm0.141}$ | $0.592_{\pm0.105}$ | $0.577_{\pm0.104}$ |
| Finetune | $0.679_{\pm0.043}$ | $0.671_{\pm0.055}$ | $0.385_{\pm0.089}$ | $0.517_{\pm0.160}$ | $0.599_{\pm0.085}$ | $0.629_{\pm0.055}$ |
| ProtoNet | $0.821_{\pm0.197}$ | $0.858_{\pm0.144}$ | $0.282_{\pm0.045}$ | $0.657_{\pm0.034}$ | $0.749_{\pm0.182}$ | $0.866_{\pm0.212}$ |
| MAML | $0.842_{\pm0.207}$ | $0.848_{\pm0.212}$ | $0.511_{\pm0.050}$ | $0.726_{\pm0.023}$ | $0.653_{\pm0.093}$ | $0.844_{\pm0.202}$ |