[Reviews · NeurIPS 2020]

Review 1

Summary and Contributions: The authors propose a novel algorithm named G-meta that applies a meta-learning algorithm to node classification or link prediction for the fast adaptation of the new task. The proposed method estimates the prototype based on the subgraph centered at each node and the entire process is optimized by using the optimization-type meta-learning algorithm, MAML. The authors also show that the information of the subgraph is rich enough to reduce the graph influence loss if the node influence between two apart nodes diminishes.

Strengths: The strength is that the authors propose a novel combination of MAML and the graph neural network that estimates the prototype constructed from local subgraph to quickly adapt a new task of the node classification task or edge prediction task, and show its effectiveness by the experiments on five datasets. Theoretical analysis is great and it also supports the validity of their approach. The validity of the proposed method is confirmed by comparing with the nine baseline methods.

Weaknesses: Although the authors show the top performance in the experiments, the experiment is not well designed to reveal which condition the proposed method works best and what part is most essential for the success. For example, meta-learning algorithm will work well if the training data in the test task is few and it is trained on the other several tasks. So if the meta-learning based method is compared with the conventional supervised learning method, it would be better to see how few the dataset in the test task the meta-learning based method can take advantage over the conventional supervised learning method when varying the size of the training dataset. Also it would be nice to see the ablation study how much the proposed method benefit from the optimization-type meta leaning algorithm by comparing the proposed algorithm with and without MAML. As for the algorithm, meta-GNN already proposed the combination of MAML and graph neural network. Meta-GNN is close to the proposed G-Meta algorithm, and the difference is the usage of the metric learning to cope with Multiple Graphs and Disjoint Labels problem. I feel the difference remains moderate.

Correctness: Although the authors claim that the proposed method covers the wider class of graph problem, and it is true when compared with some existing methods. However, it only covers node classification task and link prediction task where the target is discrete class label. It does not include the molecular prediction problems discussed in [15] that is referred in the introduction of the paper. To safely avoid the confusion, it would be better to explicitly mention the problems the proposed method solves in the paper and Figure 1.

Clarity: I think the paper is overall well written and enough clear except for the problems the proposed method solves as mentioned above.

Relation to Prior Work: Existing methods are described in the paper, but the difference with meta-Graph remains moderate.

Reproducibility: Yes

Additional Feedback:


Review 2

Summary and Contributions: This paper proposes a novel and general method for meta-learning inductive biases for graph-neural networks. The proposed method is amenable to a greater set of problems than prior methods that tend to specialise on certain cases, and show substantial improvements over prior works on several benchmarks. --- Post rebuttal update --- Thank you for a great rebuttal that addressed my main comments, especially with ablations on sub-graph size and the relative importance of metric- and gradient-based meta-learning.

Strengths: This paper is generally well-written and easy to follow. It presents a novel method for solving an important class of meta-learning problems, and demonstrates superior performance to baselines that are less general than the proposed method. As such, I recommend acceptance. In particular, the core idea of this paper is to represent each node with a local subgraph, as opposed to a full, global graph as in prior works. The degree of locality is measured by a hard cutoff threshold, and the authors motivate this locality by showing that the degree to which a node can influence another decays exponentially with distance in the graph. This suggests a bias-variance trade-off from a meta-learning point of view, and the authors results suggest that this trade-off is not trivial.

Weaknesses: My main suggestion for further improvement would be to provide a deeper analysis of some key algorithmic choices to both support claims made in the paper and help the reader understand what seems to be driving performance. At a high level, the proposed method makes use of both metric meta-learning and gradient-based meta-learning, but it is unclear how these relate and contribute to the method’s overall performance.

Correctness: Theoretical claims and empirical results appear correct to the best of my knowledge.

Clarity: This paper is well-written and easy to follow. If anything I would say that Section 3.1 feels out of place - this sits more naturally within the experimental section.

Relation to Prior Work: The related work section needs to be expanded. A single paragraph is no good. As for the choice of a meta-learned metric for node classification, this is taken from Matching Nets and Prototypical Nets, but this algorithmic choice is only briefly discussed without much of a motivation. It would strengthen the paper to discuss this choice and its relationship with prior works in greater depth.

Reproducibility: Yes

Additional Feedback: I would recommend the authors to provide an explicit ablation study to demonstrate both how sensitive the method is to the choice of this cutoff threshold and what kind of relationship we see between generalisation and subgraph locality. I would expect an inverted U-shape, where too local a subgraph would not capture sufficient information, and too large a subgraph would lead to overfitting, but this remains to be demonstrated. Similarly, it would be illuminating to see what kind of subgraphs are being extracted, in particular in the synthetic experiment.


Review 3

Summary and Contributions: The paper proposes a general framework for a variety of graph meta-learning problems. The G-Meta represents every node with a local subgraph and uses local subgraphs to transfer subgraph-specific information. Experimental results on seven datasets show promising performance, compared with several baselines. ----------------- After read all the comments from other reviews and the rebuttal of the authors. The authors addressed my concerns about ablation analysis. However, for the main concern about the subgraph, I still think there is nothing wrong with my previous understanding. Meta-GNN also computes the nodes representations using a sub-graph based on the number of aggregation layers (for l layer GNN, it computes nodes representations based on the l-hop neighbors, rather than the entire graph). Furthermore, I suppose the performance improvement of G-Meta compared with Meta-GNN comes from the usage of metric learning. Drawing conclusions from such comparisons are not fair. Therefore, I will remain my score unchanged.

Strengths: +The idea of using local subgraphs information to facilitate essential knowledge transferring via meta gradients is sound. +The theoretical analysis is substantial and reasonable. +Comprehensive experiments on seven datasets show considerable improvement. + The paper is well written in general and easy to read. The figures are illustrative.

Weaknesses: 1. The idea of using local subgraphs to compute the node representation is not novel. In practice, GCN also uses node’s neighborhood to define a computation graph and generates node embedding based on the local network neighborhoods. 2. There is a lack of experimental analysis on the influence of the different subgraph sizes on performance. 3. It seems like the baseline (ProtoNet, MAML) setting is not explained clearly. From the current description, it seems like that the ProtoNet uses the same encoder as G-Meta to get the subgraph representation h, and then uses the prototypical loss as G-Meta, which can be seen as an ablation model of G-Meta. Also, the exact setting of MAML needs to be explained clearly. 4. Comparing the performance of G-Meta and ProtoNet in Table 2, I am wondering about the reason why ProtoNet gets the comparable results on Single Graph Disjoint Labels and Multiple Graphs Disjoint Labels, but get much worse results on Multiple Graphs Shared labels.

Correctness: Yes

Clarity: Yes

Relation to Prior Work: Yes

Reproducibility: Yes

Additional Feedback: typos: 1. line 236 a varying distribution “are” -> “is” 2. line 264 use -> uses


Review 4

Summary and Contributions: --- Update after rebuttal --- I thank the authors for their detailed rebuttal, and in particular regarding the additional ablation experiments and clarifications. I maintain my recommendation to accept the paper and strongly encourage authors to integrate the additional content in the final version of their paper. --- The authors propose a generic meta-learning framework for node graph classification and link prediction when label annotations are scarce. Their key contribution is to decompose the graph inputs into a set of local subgraphs centered around the node of interest. The provide theoretical motivation that information lost by this decomposition is minimal. The approach itself combines concepts from few-shot learning prototypical networks and MAML. Main contributions are: 1- theoretical motivation for their subgraph decomposition strategy 2- applicability to a variety of settings and scalability 3- state of the art performance on multiple datasets

Strengths: - The paper is well written and motivated. In particular proofs in appendix are clear with all steps explained and justified. - The idea of exploiting only local information to learn graph representations is appealing and results suggests that it is sufficient to learn accurate representations. The increased scalability aspect is particularly appealing. Could authors comment on the impact of local decompositions on the oversmoothing problem , could this strategy alleviate the problem? - Experiments are provided on multiple datasets with state of the art performance, and show that the strategy can be exploited in multiple settings. - Authors promise to release code and datasets after the review period.

Weaknesses: - The main weakness of the work relates to the computational complexity of 1) computing the local subgraphs (are shortest paths computed ahead of the training process?), 2) evaluating each node's label individually. Can authors comment on the impact on training/evaluation time? - Another important missing element from the paper is the value of neighborhood size h, as well as an analysis of its influence over the model's performance. This is the key parameter of the proposed strategy and providing readers with intuitive knowledge of the value of h to use, and the robustness of the method with respect to larger or smaller neighborhoods is essential. Similarly, different hyperparameter sets are used per dataset, which is not ideal. Can authors provide insights into how performance varies with a constant set of parameters? - Certain aspects of the training set-up needs clarifying. Mainly the task generation process (what constitutes a task, can one task contain multiple graphs, are local substructures randomly sampled regardless of the original graph, are all nodes labelled in the training set, etc) - Certain sections are too condensed and would be much clearer and informative if expanded (e.g. related work, training setup, testing setup, baseline methods). In the interest of space, table 1 could be moved to supplementary.

Correctness: I have not seen issues in the proofs provided in supplementary. A few comments claims can be clarified in the results sections, e.g. l 276, please specify the setting/dataset and which baseline model is referred to (e.g KNN model or meta-GNN type methods). - Can authors please comment on the claim that methods typically require large amounts of annotated data and clarify the distinction between the studied set-up and the standard semi-supervised setting when only a small fraction of nodes is labelled ?

Clarity: The paper is for the most part clear and well written, albeit a little condensed at times. As mentioned in the weaknesses section, the experimental set-up needs additional attention: -Are all graph nodes labelled in the training set? Are K shot settings simulated as in standard episode training? -How are tasks constructed, especially with respect to the settings with multiple graphs? -Can authors provide a clear description of the testing setting? In particular with respect to the query set structure, and evaluation: is each node labeled individually? -When working in 5-shot settings, is it 5 nodes across 5 graphs? Regarding baseline methods compared to, am I correct to assume that this constitutes some form of ablative experiment? If so, please highlight that this is a variant of the proposed work and highlight the main difference with the complete method.

Relation to Prior Work: The related work section is very condensed but reviews main works and main differences. The paragraph at line 257 provides an additional description of the most similar methods and compares to main works exploring few-shot/meta-learning on graphs.

Reproducibility: Yes

Additional Feedback: l. 104: it is mentioned that the label set size over the whole data set is N, how does that relate to M? Considering it is mentioned next that this constitutes a N-way K shot setting, does this mean that N is the number of labels per task? If so, please clarify the distinction in the text. l. 142: please briefly clarify the notation x^(inf). l. 147: the proof makes the assumption that graph weights are non negative, this should be mentioned here. l. 154: neighborhood size is often referred to as the number of nodes in the neighborhood of a given nodes, I would suggest to make the distinction clear here that h is the number of hops from node u.

[Author Response · NeurIPS 2020]

We thank the reviewers for their time and valuable feedback. Overall, reviewers found our method G-META *"novel"*,
*"scalability is particularly appealing"*, *"theoretical analysis is great, substantial, valid, correct"*, experiments are
*"comprehensive"*, and the paper *"well-written"*. Below, we clarify important points raised by reviewers: **(1) relation to**
**existing work**, **(2) local subgraphs in G-META vs. computation graphs in GNNs**, **(3) size of local subgraphs**, **(4)**
**ablation**, and **(5) others.** We believe these clarifications, together with our new analyses, resolve all key issues raised.

**(1) Relation to existing work.** **R1** questioned the difference between Meta-GNN and G-META, saying it *"remains*
*moderate."* We respectfully disagree. The key difference are local subgraphs, and this is crucial because local subgraphs
enable i) accurate and fast adaptation to new tasks, ii) learning in few-shot settings, and iii) theoretical justification. In
contrast to our G-META, Meta-GNN and, similarly, Meta-Graph, use *entire graphs* for meta-learning. Because of that,
they are unable to propagate label information across large graphs when only a few node labels are given. Theoretically,
Meta-GNN/Meta-Graph works only on 1 out of 3 graph meta-learning problems (see Appendix B) and does not have any
theoretical motivation whereas G-META works on all 3 problems and is theoretically justified. Empirically, G-META
outperforms Meta-GNN, for example, by 65% on the ogbn-arxiv dataset (Meta-GNN cannot even be used on other
datasets and meta-learning regimes). As suggested by reviewers, we will carefully discuss this in the final version.

**(2) Local subgraphs vs. computation graphs.** **R3** raised a critical concern that *"the idea of using local subgraphs*
*to compute node representations is not novel."* This points to a **critical misunderstanding—computation graphs in**
**GNNs are used to generate node embeddings vs. local subgraphs in our G-META are used to transfer knowledge**
**for graph meta-learning.** We are not claiming novelty in *"the idea of using local subgraphs to compute node*
*representations."* Instead, as we write in the paper, we are claiming novelty in the idea of using local subgraphs for graph
meta-learning, as recognized by **R1** and **R2**. This innovation has important implications, which we show theoretically
(i.e., proofs, solving classes of graph meta-learning problems not solved before) and empirically (i.e., considerable
boost in accuracy over 9 baselines and 7 datasets). For example, a baseline method Meta-GNN, which uses a standard
GNN on an entire graph together with MAML, performs 42.5% worse than G-META-MAML, a simplified variant of
our G-META. Note that the only difference between Meta-GNN and this simplified G-META's variant is that it uses the
entire graph vs. local subgraphs. We will clearly mention this contribution, which we agree is crucial for G-META.

**(3) Size of local subgraphs and computational complexity.** (3.1) **R2**, **R3**, and **R4**
nicely point that G-META's performance can vary with local subgraph size $h$. To address
this, we evaluate G-META for $h = 1, 2, 3$ (Figure for Fold-PPI). We find that $h = 2$ gives
the best performance, which nicely corroborates **R2**'s hypothesis on the inverted U-shape
relationship. We will include this analysis in our final version. (3.2) **R4** also raises an
important concern on computational complexity of subgraph extraction. We would like to

clarify that we don't need to compute *"shortest paths ahead of the training"*. Instead, we simply do a lookup, retrieving
neighbors of neighbors. Empirically, we find the subgraph construction takes 14.7% of training time, and this can be
further reduced by implementing techniques for improving training efficiency, e.g., GraphSAINT [Zeng et al., ICLR
2020]. Regarding **R4**'s comment of *"evaluating each node label individually"*, we note that each mini-batch consists
only of a few labels (e.g., 9 in 3-way 3-shot learning) and as such is a cheap operation.

**(4) Further ablations.** **All reviewers** raise an important point on ablation, which
we agree is crucial especially for G-META. While we already included ablation
in the form of baselines, we will make it explicit in the next version following
**R3** and **R4**. We thank **R3** and **R4** for nicely pointing out that baselines ProtoNet

| Method | ogbn-arxiv | Tissue-PPI | Fold-PPI |
|---|---|---|---|
| G-META | **0.451** | **0.768** | **0.561** |
| - MAML | 0.372 | 0.546 | 0.382 |
| - Prototype | 0.389 | 0.745 | 0.482 |

and MAML can be seen as G-META's ablations. In response to **R2**, new results in Table show that gradient-based meta-
learning aids G-META more than metric-based meta-learning but both are indispensable for G-META's performance.

**(5) Baselines and further clarifications. (5.1)** **R1** raised an important point about performance under varying size of
the training set, i.e., $K$ in the $K$-shot problem. We conduct experiments and observe, as expected, a linear trend between
$K$ and performance (e.g., on arxiv-ogbn, accuracy goes from 0.373, 0.442 to 0.484 for $K = 1, 3$, and 10, respectively).
We will include the full study in the final version. **(5.2)** **R1** correctly pointed out G-META *"covers node classification*
*task and link prediction task where the target is discrete class label.".* In contrast, existing Meta-GNN and Meta-Graph
only work for one of these two tasks—G-META is the first to work on both tasks. Also, to avoid confusion, we will
update reference for graph-level molecular prediction [Hu et al., ICLR 2019] to edge-level interaction prediction [Zitnik
et al., Bioinf. 2018]. **(5.3)** We thank **R4** for rightly pointing out that local subgraphs can alleviate over-smoothing
because, in each iteration, different subgraphs are fed into GNN, which promotes inductive generalization. **(5.4)** **R4**
raised a question about hyper-parameters. We use random search on validation set to select hyper-parameters and
find that model performance is stable for a broad range of values. We will include the recommended set of constant
parameters in the next version. **(5.5)** **R4** raised an important point on experimental setup. We follow standard episode
training and semi-supervised setting in which most nodes are not labeled, i.e., few-shot learning. In $K$-shot $N$-way
setup, for "Multiple Graphs and Shared Labels" problem, each task samples $K$ nodes for each label $N_i$ in the same
label set of size $N$ from one graph, and different tasks are associated with different graphs. For "Multiple Graph and
Disjoint Labels" problem, each task defines an $N$-size label set, and samples $K$ nodes for each label $N_i$.

[Meta-Review · NeurIPS 2020]

This paper proposes a meta-learning method for graph data. The use of the local subgraphs provides more flexibility and allows us to adopt the same framework to different scenarios. The effectiveness of the proposed method is supported by theory and experiments. However, Meta-GNN also computes the nodes representations using a sub-graph based on the number of aggregation layers. Also, the performance improvement of the proposed method compared with Meta-GNN might come from the usage of metric learning. The paper should be revised such that the advantages of the proposed method over Meta-GNN become clearer.